# Deep Learning-Based Total Kidney Volume Segmentation in Autosomal Dominant Polycystic Kidney Disease Using Attention, Cosine Loss, and Sharpness Aware Minimization

**DOI:** 10.3390/diagnostics12051159

**Published:** 2022-05-07

**Authors:** Anish Raj, Fabian Tollens, Laura Hansen, Alena-Kathrin Golla, Lothar R. Schad, Dominik Nörenberg, Frank G. Zöllner

**Affiliations:** 1Computer Assisted Clinical Medicine, Mannheim Institute for Intelligent Systems in Medicine, Medical Faculty Mannheim, Heidelberg University, 68167 Mannheim, Germany; laura.hansen@medma.uni-heidelberg.de (L.H.); alena-kathrin.golla@medma.uni-heidelberg.de (A.-K.G.); lothar.schad@medma.uni-heidelberg.de (L.R.S.); frank.zoellner@medma.uni-heidelberg.de (F.G.Z.); 2Department of Radiology and Nuclear Medicine, University Medical Centre Mannheim, Medical Faculty Mannheim, Heidelberg University, 68167 Mannheim, Germany; fabian.tollens@medma.uni-heidelberg.de (F.T.); dominik.noerenberg@medma.uni-heidelberg.de (D.N.)

**Keywords:** attention, autosomal dominant polycystic kidney disease, cosine loss, deep learning, image segmentation, magnetic resonance imaging, sharpness aware minimization, total kidney volume

## Abstract

Early detection of the autosomal dominant polycystic kidney disease (ADPKD) is crucial as it is one of the most common causes of end-stage renal disease (ESRD) and kidney failure. The total kidney volume (TKV) can be used as a biomarker to quantify disease progression. The TKV calculation requires accurate delineation of kidney volumes, which is usually performed manually by an expert physician. However, this is time-consuming and automated segmentation is warranted. Furthermore, the scarcity of large annotated datasets hinders the development of deep learning solutions. In this work, we address this problem by implementing three attention mechanisms into the U-Net to improve TKV estimation. Additionally, we implement a cosine loss function that works well on image classification tasks with small datasets. Lastly, we apply a technique called sharpness aware minimization (SAM) that helps improve the generalizability of networks. Our results show significant improvements (*p*-value < 0.05) over the reference kidney segmentation U-Net. We show that the attention mechanisms and/or the cosine loss with SAM can achieve a dice score (DSC) of 0.918, a mean symmetric surface distance (MSSD) of 1.20 mm with the mean TKV difference of −1.72%, and R${}^{2}$ of 0.96 while using only 100 MRI datasets for training and testing. Furthermore, we tested four ensembles and obtained improvements over the best individual network, achieving a DSC and MSSD of 0.922 and 1.09 mm, respectively.

## 1. Introduction

Autosomal dominant polycystic kidney disease (ADPKD) is a hereditary disorder with a slow and gradual development of cysts in the kidneys. ADPKD leads to renal enlargement and eventually to end-stage renal disease (ESRD) with renal failure [1,2]. Daalgard et al. [3] reported that ESRD might occur within five years after detecting ADPKD. Hence, it is important to monitor ADPKD progression in patients. The total kidney volume (TKV) increases with ADPKD progression and, therefore, can be used as a risk predictor of disease development [4] and to quantify disease progression [5].

Consequently, the Food and Drug Administration (FDA) has accepted TKV as an important biomarker [6] for determining renal function in patients. Moreover, the TKV is the only MRI-established biomarker so far [7] and can be determined by segmenting the kidneys from the MRI volumes. Manual segmentation by an experienced physician is, however, time-consuming and prone to observer variability [8]. Alternatively, deep learning approaches are generally much faster and have recently achieved state-of-the-art results in medical imaging [9].

In the past, kidney segmentation has been approached via classical image processing techniques, such as algorithmic segmentation methods, model-based segmentation methods, and their combinations. An overview of related work is given in two reviews by Zöllner et al. [7,8]. However, only a few machine learning (especially deep learning approaches) for kidney segmentation have recently been proposed.

Kline et al. [10] approached kidney segmentation of ADPKD patients with 2000 training and 400 test cases. They employed multi-observer artificial neural networks consisting of 11 CNNs with variable depths and parameters. They post-processed the prediction map using the two largest connected components and then applied active contours and edge detection to finalize the segmentation. The resulting average dice score (DSC) was 0.97. Based on this work, van Gastel et al. [11] used T2-weighted MR images from 440 ADPKD patients for training and extended the method to also include liver segmentation. They added inception blocks and residual connections to the network in [10], obtaining a DSC of 0.96. In another approach, Bevilacqua et al. [12] implemented R-CNN to first determine the ROI containing kidneys and then a semantic segmentation CNN to delineate the kidneys. Their approach reached a DSC of 0.88 with a dataset of 57 images from four patients. Mu et al. [13] employed a multi-resolution method using a modified V-Net [14] to segment ADPKD kidneys in 305 patients. The resulting DSC reached 0.95. Daniel et al. [15] developed an automated system using the U-Net [16] to segment kidneys and ultimately determine TKV for renal disease detection. They used T2-weighted MR images from 30 healthy and 30 chronic kidney disease (CKD) patients. Their system achieved a DSC of 0.93 on a test dataset consisting of 10 patients.

The drawback of such deep learning approaches is that they require huge amounts of data to train the networks to achieve high (segmentation) accuracy. This problem is further escalated in the medical sector where image data are rare in most cases. The reason is that acquiring large medical datasets is hindered by the complexity and high cost of large-scale experiments or in the case of a rare disease, a limited number of patients. Additionally, a class imbalance between the background and the segmented object exists. To mitigate this problem, data augmentation techniques [17] or approaches to generate synthetic data are proposed [18,19].

Our contribution in this work is to address the problem of a small dataset by introducing a cosine loss function, which, to the best of our knowledge, has not been implemented so far for medical image segmentation tasks. Moreover, we integrate sharpness aware minimization (SAM) [20] with the loss function to improve model generalizability. We further investigate and incorporate three attention mechanisms [21,22,23,24,25] with convolutional neural networks (CNNs) so that the networks can focus on relevant image regions. As a final experiment, we explore four ensembles that consist of different attention networks and loss functions with SAM for automatic kidney volume segmentation in patients with ADPKD.

## 2. Materials and Methods

### 2.1. Image Data

The patient image data were obtained from the National Institute of Diabetes and Digestive and Kidney Disease (NIDDK), National Institutes of Health, USA, and were recorded in the Consortium for Radiologic Imaging Studies of Polycystic Kidney Disease (CRISP) study [2]. It contains T1- and T2-weighted MRI scans of patients with different stages of CKD. For this work, we retrieved 100 datasets from the NIDDK database. The male-to-female ratio is 50:50 with an average age of 30±10 years. It includes patients from healthy (CKD stage 1) to ADPKD (CKD stage 3) cases. The number of cases belonging to CKD stages 1, 2, and 3 is 41, 41, and 18, respectively. We only focused on T1-weighted MRIs. Images were recorded with a matrix of 256 × 256 and 30–80 slices with an in-plane resolution of 1.41 × 1.41 (±0.13) mm${}^{2}$ and slice thickness of 3.06 (±0.29) mm. Images were recorded in the coronal orientation. More details of the imaging protocol can be found in the original study in [2].

### 2.2. Image Annotation

For each dataset, left and right kidneys including cysts, were segmented manually as a reference standard. Two experienced physicians independently performed segmentation on coronal MR images using an in-house developed annotation tool based on MeVisLab SDK (MeVis Medical Solutions, Inc., Bremen, Germany), which also allowed for an analysis of the inter-user agreement of kidney segmentations. The mean inter-user agreement was found to be 0.91 ± 0.06 (Dice) with a coefficient of variation of 0.07.

### 2.3. Pre-Processing

We first normalized the images using Equation (Equation 1),
(1)$$\hat{I}=\frac{I-\mu (I)}{\sigma (I)},$$
where μ(I) and σ(I) are the mean and standard deviations, respectively, of the original image *I*. $\hat{I}$ is the normalized image. Furthermore, as a data augmentation technique, we used a constrained label sample mining approach where patches were extracted from MRI slices with patch center probability of 50:50 on the label:background pixel for each batch [26]. We trained the networks on patches of size 96 × 96 and 128 × 128. For testing, we used the whole image size of 256 × 256. The pre-processing was implemented using SimpleITK 1.2.4 [27].

### 2.4. Attention Module

Oktay et al. [21] proposed attention gates in the U-Net to guide the network in selecting relevant features and disregard irrelevant ones by using higher-level features as a guide to suppress trivial and noisy responses in the lower-level skip connections. Figure 1 illustrates this attention module.

The gating signal *g* is the higher-level feature and $x^{l}$ is the corresponding previous layer skip connection. Since *g* is spatially smaller and has more feature maps than $x^{l}$, it is convoluted with a 1 × 1 filter to obtain the same number of features. Afterward, *g* is upsampled to have the same spatial dimensions as $x^{l}$. Then, both *g* and $x^{l}$ are concatenated followed by ReLU, 1 × 1 convolution, sigmoid activation, and eventually an upsampling layer resulting in the attention coefficients α. The attention coefficients are then multiplied with the skip connection to obtain the final attention feature maps.

### 2.5. Convolutional Block Attention Module

The convolutional block attention module (CBAM) calculates and combines both spatial and channel attention into one network [23]. Briefly, a 1D channel attention map and thereafter a 2D spatial attention map are computed. The spatial attention map is generated by performing max and average-pooling operations along the channel dimension of the input feature map. The pooled feature maps are concatenated and fed forward through a convolution layer to yield a 2D spatial attention map. There are two versions of channel attention: (1) squeeze and excitation attention proposed by Hu et al. [22] and (2) channel attention, as outlined in [23]:

#### 2.5.1. Squeeze and Excitation Attention

This attention mechanism focuses on channel relationships in a network. Briefly, the attention module performs two operations sequentially to form a squeeze and excitation (SE) block. First, a squeeze operation on the input layer is executed, which is then followed by an excitation operation. The squeeze operator does a global average-pooling of spatial information into a channel descriptor of size 1 × 1 × C, where C is the number of channels in the input layer. This output vector is then processed by an excitation operation that captures full channel-wise dependencies [22]. The following equation describes the exact excitation operation,
(2)$$E_{x}=\phi_{sig}\left(W_{2}\phi_{ReLU}\left(W_{1}z\right)\right),$$
where z is the output from the squeeze operation, $\phi_{ReLU}$ the ReLU activation, $W_{1}$ & $W_{2}$ are two fully connected (FC) layers, and $\phi_{sig}$ is the sigmoid activation. The final output is obtained by multiplication of $E_{x}$ and the input layer. The squeeze and excitation operation is depicted in Figure 2.

#### 2.5.2. Channel Attention in CBAM

The CBAM’s channel attention [23] is calculated by squeezing the spatial dimension of the input feature map using average and max-pooling and then feeding them forward through a small multi-layer perceptron (MLP).

The channel and spatial attention modules used in CBAM are illustrated in Figure 3a. The complete CBAM block is shown in Figure 3b. In our experiments, we incorporate SE and CBAM blocks in the Attention U-Net [21] encoder to implement the SE U-Net and CBAM U-Net, respectively.

### 2.6. Cosine Loss

The cosine loss has been shown to improve the image classification accuracy for small datasets [28]. Hence, we adapt this loss function for our kidney segmentation task. The loss is given by,
(3)$$S\left(\hat{Y},Y\right)=\frac{\langle \hat{Y},Y\rangle }{{∥\hat{Y}∥}_{2}\cdot {∥Y∥}_{2}},$$
(4)$$L_{COS}\left(\hat{Y},Y\right)=1-S\left(\hat{Y},Y\right),$$
where *S* and $L_{COS}$ are the cosine similarity and cosine loss, respectively, between the prediction $\hat{Y}$ and the ground truth *Y*.

### 2.7. Sharpness Aware Minimization

Foret et al. [20] introduced a technique called sharpness aware minimization (SAM) that helps improve the generalizability of neural networks. Briefly, the method searches for a neighborhood of parameters with homogeneous low loss values, signifying a wide loss curve at the minimum, thereby, reducing the loss value and sharpness of the loss curve. A wide minimum suggests that the parameters in the neighborhood will generally yield consistently better predictions compared to a minimum with a sharp curve.

### 2.8. Networks

All networks implemented in this work are based on the 2D U-Net architecture by Ronneberger et al. [16] extended with residual connections. The baseline 2D U-Net architecture is illustrated in Figure 4a. For the baseline experiments, the 2D U-Net is used with DSC loss (cf. Equation (Equation 5)) and a combination of DSC and cross-entropy (CE) loss.
(5)$$L_{DSC}\left(\hat{Y},Y\right)=1-\frac{2\cdot \sum_{c=1}^{3}\sum_{i=1}^{N}{\hat{y}}_{i,c}\cdot y_{i,c}}{\sum_{c=1}^{3}\sum_{i=1}^{N}{\hat{y}}_{i,c}+\sum_{c=1}^{3}\sum_{i=1}^{N}y_{i,c}}$$

The cross-entropy loss is given by,
(6)$$L_{CE}\left(\hat{Y},Y\right)=-\frac{\sum_{c=1}^{3}\sum_{i=1}^{N}y_{i,c}\;log\left({\hat{y}}_{i,c}\right)}{3\cdot N}$$
where ${\hat{y}}_{i,c}$ and $y_{i,c}$ correspond to the individual voxel probabilities and label, respectively, with *c* and *N* being the number of classes and voxels in a batch, respectively. Equation (Equation 7) depicts the combination of $L_{CE}$ and $L_{DSC}$ where $L_{CE}$ is weighted with λ=10 [29].
(7)$$\begin{array}{cc} L_{CE+DSC}\left(\hat{Y},Y\right) & =L_{DSC}\left(\hat{Y},Y\right)+\lambda \;L_{CE}\left(\hat{Y},Y\right) \end{array}$$

The Attention U-Net as introduced in Section 2.4, is depicted in Figure 4b. The other variants that involve SE modules/CBAMs in the Attention U-Net encoder part are shown in Figure 4c. In preliminary experiments, we found that modifying baseline U-Net with only CBAMs or SE modules performed worse or similar to the baseline. Hence, for any further experiments, we combined them only with the Attention U-Net. All described networks were implemented using TensorFlow 2.0 and Python 3.7.

### 2.9. Training

We trained the networks on T1-weighted MR images and implemented a batch size of 16 and 8 for patch sizes 96 and 128, respectively, using the Adam optimizer [30] and a learning rate of $10^{-3}$. We used exponential linear units (elus) [31] as activation functions with batch normalization, L2-regularization ($10^{-7}$) and drop-out with probability of 0.01. Furthermore, we performed 5-fold cross-validation with a split of 70:10:20 patient image volumes in train:validation:test sets. We selected 160 samples per patient MRI volume during training.

Each network was trained for at least 20 epochs. After that, the training stopped as soon as the difference in segmentation accuracy of each kidney in the validation data were less than $10^{-4}$ over the last ten epochs. We then selected the network’s weights with the highest average accuracy on the validation data from these last ten epochs.

### 2.10. Ensembles

We created two ensembles from our proposed networks. The first one consists of four networks (SE U-Net, CBAM U-Net, Attention U-Net, and U-Net) trained using cosine loss with SAM ($L_{\mathrm{COS}}+\mathrm{SAM}$). The second ensemble consists of seven networks that include the four networks from the first ensemble plus three networks (SE U-Net, CBAM U-Net, and Attention U-Net) trained using cross-entropy + DSC loss with SAM ($L_{\mathrm{CE}+\mathrm{DSC}}+\mathrm{SAM}$). We selected these networks to test all attention networks and loss functions. We employed two methods for calculating the ensemble result: (1) simultaneous truth and performance level estimation (STAPLE) [32] and (2) majority voting.

### 2.11. Evaluation

To compare the proposed models, we used the DSC, the mean symmetric surface distance (MSSD), and the TKV as evaluation metrics. Firstly, we used the DSC similarity coefficient to assess the overlap between the ground truth *Y* and the segmentation $\hat{Y}$,
(8)$$DSC\left(\hat{Y},Y\right)=\frac{2\;|\hat{Y}\cap Y|}{|\hat{Y}\left|+|Y\right|}$$

Secondly, we employed the MSSD (in mm) that is more perceptive to alignment and shape:(9)$$MSSD\left(\hat{Y},Y\right)=\frac{\sum_{\hat{y}\in \hat{Y}}\min d(\hat{y},Y)+\sum_{y\in Y}\min d(y,\hat{Y})}{|\hat{Y}\left|+|Y\right|}$$

Finally, we calculated the TKV (in mL) by multiplying the number of voxels belonging to the segmented kidneys by their voxel volume (mm${}^{3}$) divided by 1000 to convert the result to mL.

We compared the TKVs of the manual and the obtained segmented kidneys of our networks using scatter plots and the coefficient of determination (R${}^{2}$).

Furthermore, we used a paired t-test to check for the significance between the results from the baseline and our implemented methods. Here, the null hypothesis is that the baseline network configuration is better than the developed methods for the given evaluation metric. It is rejected at p<0.05.

## 3. Results

The quantitative results for all experiments are displayed in Table 1 where the DSC and MSSD values are averaged over both the kidneys for better comprehension. For the baseline U-Net (Figure 4a), the DSC loss ($L_{DSC}$) performs worse for all metrics and patch sizes than the combination of cross-entropy and DSC loss ($L_{CE+DSC}$). Furthermore, all proposed methods perform better than the baseline. Post-processing employing the largest connected components further improves the DSC and MSSD. In most cases, improvements are significant (see Table 2). The best results among individual networks were obtained using the U-Net with $L_{COS}+SAM$ and patch size of 128. Here, an average DSC of 0.918±0.044 and an MSSD of 1.199±1.525 mm were achieved. The segmentation quality was further improved using ensembles. The DSC and MSSD of 0.922±0.047 and 1.094±1.376 mm, respectively, were achieved using the ensemble of seven models with the majority voting scheme. Table A1 displays results for the left and right kidneys.

Examples of obtained segmentation after post-processing for each network are displayed in Figure 5. The top row depicts a case with a DSC of 0.96 (stage: 1, female, age: 34), where the two networks (U-Net ($L_{\mathrm{COS}}$) & Attention U-Net ($L_{\mathrm{CE}+\mathrm{DSC}}$)) surpass the baseline only by a DSC of ≈ 0.002. The lower row depicts the obtained segmentation from a case with a high load of cysts that are distributed not only in the kidneys but all over the abdomen (stage: 2, female, age: 32). We observe that every network has difficulties obtaining a DSC > 0.80. Nonetheless, the proposed networks outperform the baseline U-Net by up to 18% (U-Net ($L_{\mathrm{COS}}+\mathrm{SAM}$)).

Investigating the cases with DSC ≤0.80, we observe that our proposed networks can reduce the number of such cases by half: 13 for the baseline U-Net ($L_{CE+DSC}$) versus 5 for the other networks. It is also worth noting that these 5 cases were among the 13 cases with low DSC of the baseline U-Net. Moreover, the ensemble with 7 networks reduces this number to only 3 cases.

### 3.1. Attention Mechanisms

The networks SE U-Net, CBAM U-Net, and the Attention U-Net with $L_{CE+DSC}$ as loss function all outperform the baseline results across all metrics (Table 1). For CBAM and Attention U-Net with $L_{CE+DSC}$, the results are significantly better (*p*-value < 0.05) than the baseline U-Net ($L_{CE+DSC}$) for 3/4 metrics (both the DSC scores and one MSSD value). In this configuration, the SE U-Net outperforms the other two for patch size of 96, however, for patch size of 128, CBAM and Attention U-Net perform better in terms of the DSC (0.899 and 0.898, respectively).

Within the test set, five cases had a DSC ≤0.8. The average DSC of these five cases in baseline U-Net ($L_{CE+DSC}$) is 0.638 ± 0.081, while for the Attention U-Net ($L_{CE+DSC}$), this score rises by 6% to 0.704 ± 0.072.

### 3.2. Cosine Loss

Meanwhile, for the cosine loss ($L_{COS}$), we again observe that the U-Net, the Attention U-Net, and the CBAM U-Net outperform the baselines significantly (*p*-value < 0.05) over all the patch sizes and metrics. We also find that U-Net and CBAM U-Net with $L_{COS}$ provide the best DSC and MSSD values for patch sizes 128 and 96, respectively, among the networks without SAM. Furthermore, the Attention U-Net with $L_{COS}$ surpasses its corresponding network with $L_{CE+DSC}$ over both the DSC and one MSSD value (patch: 128).

For the Attention U-Net ($L_{COS}$) an improvement of 3% in DSC is recorded (from DSC of 0.647 to 0.677) as compared to the baseline U-Net for the five cases with DSC ≤0.8.

### 3.3. SAM

The application of SAM on top of the networks yields the best performance overall for the individual networks. The SE U-Net with $L_{COS}+SAM$ provides the best DSC (0.902) and MSSD (2.228 mm) values for patch size 96. Furthermore, the U-Net with $L_{COS}+SAM$ yields the best DSC of 0.909 with patch size 128 among all individual networks. Moreover, the overall best MSSD value of 2.09 mm is provided by the CBAM U-Net with $L_{COS}+SAM$. We further observe that in six out of eight cases, the $L_{COS}+SAM$ performs better than the corresponding $L_{CE+DSC}+SAM$ when the respective DSC and MSSD values are compared.

For cases with DSC ≤0.8, the Attention U-Net with $L_{COS}+SAM$ produces similar segmentation results as described before. Again, the network surpasses the baseline U-Net.

### 3.4. Ensemble

We observe that every ensemble model achieves a higher DSC than any individual network for the same patch size. The same is observed for MSSD values except for the ensembles with STAPLE and the patch size of 96. The highest results are obtained using ensemble with seven models and majority voting (DSC: 0.918±0.048 & MSSD: 1.484±2.083 mm, see Table 1). The DSC and MSSD after post-processing reach 0.922 ± 0.047 and 1.094 ± 1.376 mm, respectively. We also observe that some ensemble results are significantly better (*p*-value < 0.05) than the corresponding best individual network (marked in italic in Table 1 and Table 2). Moreover, the ensembles with majority voting outperform those using STAPLE. No significant differences in performance between the ensembles with four and seven models were observed (p>0.05).

### 3.5. Evaluation of Total Kidney Volume

Table 3 displays the R${}^{2}$ values of manual segmented TKVs (ground truth) versus calculated TKVs from the selected networks’ segmentations. We observe that there is a high correlation between ground truth and segmentation for smaller volumes, while for larger volumes over- or under segmentation occurs. The R${}^{2}$ for all networks is greater than 0.91, supporting the visual analysis. Furthermore, all the networks except one (U-Net ($L_{COS}$)) outperform the baseline demonstrated by a higher R${}^{2}$ with less deviation in the mean TKV difference (%) values. This conforms to increased segmentation accuracy in DSC and MSSD (cf. Table 2). The highest R${}^{2}$ value of 0.9626 is achieved by the Attention U-Net ($L_{CE+DSC}$). It outperforms the baseline U-Net by 5%. Furthermore, the mean TKV difference for the baseline is −4.43 ± 18.9%. In comparison, the mean TKV difference for the Attention U-Net ($L_{CE+DSC}$) and the U-Net ($L_{COS}+SAM$) are −2.04 ± 12.2% and −1.72 ± 12.5%, respectively. The mean TKV differences for the ensembles of 7 networks and majority voting or STAPLE are −0.65 ± 13.76% and −4.00 ± 14.82%, respectively. Meanwhile, the mean TKV differences for four network ensembles with majority voting or STAPLE are 2.34 ± 13.36% and −3.11 ± 14.63%, respectively.

## 4. Discussion

In this work, we investigated the impact of various attention modules, cosine loss, and SAM for improving kidney segmentation in ADPKD from T1-weighted MR images to estimate TKV while mitigating the problem of limited data. Thereby, our goal was to achieve high segmentation accuracy while only using a limited number of annotated data. Compared to other approaches reported in the literature [10,11], we achieved similar results but using only a fraction of the data employed by others. We further conducted experiments with ensembles of our networks yielding more improvements. In the following, we first discuss the impact of individual networks and then our results combining these into ensembles.

### 4.1. Individual Networks

We found that all attention networks with $L_{\mathrm{COS}}$ and $L_{\mathrm{CE}+\mathrm{DSC}}$ outperformed the baseline networks across all metrics (Table 1). Combining the baseline U-Net with $L_{\mathrm{COS}}$ and SAM yielded the best result among the individual networks with DSC scores of 0.918 ± 0.04 and 0.908 ± 0.05 for patch sizes 128 and 96, respectively. Additionally, the MSSD was minimal for this combination. Generally, we observed that patch size can play an important role. Here, most of the networks that have been trained with a patch size of 128 significantly outperformed the networks trained with a patch size of 96. For example, the U-Net ($L_{\mathrm{COS}}+\mathrm{SAM}$) with a patch size of 128 significantly outperformed the same U-Net with a patch size of 96 (*p* = 0.0008 for DSC, *p* = 0.039 for MSSD).

The impact of the Attention U-Net is more prominent for challenging cases (i.e., segmentations with DSC < 0.80). The number of such cases for the baseline U-Net ($L_{\mathrm{CE}+\mathrm{DSC}}$) was 13; however, this number is reduced to 5 in the case of the proposed networks. These five cases were common to both networks, indicating the failure of the U-Net architecture to accurately segment such difficult samples. These samples contain cysts in various regions of the abdomen and less-defined kidney boundaries and shapes.

Even though these cases were difficult to segment by all networks, we still see an improvement of up to 6% in the DSC compared to the baseline U-Net from DSC = 0.638 to the Attention U-Net ($L_{\mathrm{CE}+\mathrm{DSC}}$) with DSC = 0.704. A reason might be that the U-Net ($L_{\mathrm{CE}+\mathrm{DSC}}$) is unable to focus on relevant image information, e.g., kidney boundaries with low contrast for such difficult cases. However, the attention mechanism [21] uses higher-level features as a guide to help lower-level features to emphasize such regions in the image. The Attention, CBAM, and SE U-Nets were significantly better (*p*-value < 0.05) than the baselines, suggesting that the three attention mechanisms can be integrated to improve performance.

Besides the attention mechanisms, we found that simply training the U-Net with cosine loss significantly improved the performance (*p*-value <0.05) over every metric (see Table 1). Every voxel segmented by the U-Net with $L_{\mathrm{COS}}$ lies on a unit hypersphere as they are l2-normalized. This way, the wrongly segmented voxels are heavily penalized by the cosine loss leading to the adaption of the weights during training. Results by Payer et al. support our results, though their loss function [33] is different to the one described in [28], which is implemented here. Nevertheless, our results show that the implemented loss function is also suitable for the medical image segmentation task at hand. Similar results using the cosine distance function in k-means clustering of DCE-MRI for kidney segmentation have been reported [34], outperforming standard cluster similarity metrics. In another work [35], the authors used cosine similarity and attained state-of-the-art semantic segmentation results on the following datasets: ADE20K [36], Cityscapes [37], and COCO-Stuff [38]. In conclusion, the cosine loss function can be valuable in renal MRI segmentation.

The combination of SAM with our network architectures further boosts the performance. This indicates the usefulness of SAM in improving the generalizability of the models. However, a drawback is that it takes about twice the amount of time to train such a network compared to the same network without SAM. The reason is the need to calculate gradients twice in each iteration as it first calculates gradients for the weights and then for the neighborhood parameters. Nonetheless, this limitation renders minor with respect to the steadily increasing computational power available.

Furthermore, we notice that, on average, cosine loss (without SAM) brings about 0.65% improvement in DSC over the corresponding baseline as compared to 0.53% when only using SAM. This shows that the cosine loss is more important for segmentation accuracy than SAM. However, we also find that combining cosine loss with SAM yields an average DSC improvement of 1.35%. Hence, the combination of the two is vital for attaining best segmentation accuracy.

In comparison to other deep learning-based kidney segmentation approaches, our methods perform similarly. The approaches by Kline et al. [10], van Gastel et al. [11], and Mu et al. [13] report higher DSC (up to 0.97). Consequently, their mean TKV difference is smaller than our proposed method. However, these studies use larger datasets (up to 2400). Nonetheless, in our work, the aim was to explore and combine techniques that could deal with limited data. In this respect, using only a fraction of the data (100 cases), we still achieved comparable results. Increasing the number of datasets for our method might further improve the results. However, we explicitly aimed at investigating techniques mitigating limited data. Therefore, such a test is beyond the aim of this study.

Daniel et al. [15] attained slightly better DSC (0.93) than our methods using a plain U-Net on T2 images. They also used a small dataset for training and testing. However, the major difference is that they applied their approach to healthy volunteers and patients with chronic kidney disease. Neither of these data contained cysts that altered the appearance of the kidneys in the MRI drastically (see Figure 5). Furthermore, they trained their network on T2-weighted images while we conducted our study on T1-weighted images, which could also be a reason for the different performance. Finally, we outperformed the approach in [12] with a margin of 4%; however, they used data only from four patients.

### 4.2. Ensembles

Creating ensembles of our proposed networks further improved segmentation accuracy compared to the best performing individual network. The networks combined in an ensemble are known to reduce the variance component of the prediction error and, therefore, smooth out the predictions [39,40]. The ensemble with seven networks and majority voting achieved the highest DSC and MSSD of 0.922 ± 0.047 and 1.094 ± 1.376 mm, respectively (Table 2). No significant differences in performances of ensembles with four and seven networks could be observed (*p*-value > 0.05, Table 1).

Our ensembles cannot outperform the ones presented by Kline et al. [10]. However, the key differences lie in the data size and the number of networks employed. Kline et al. used a 24-fold larger dataset and an ensemble of 11 networks. Considering this, we believe our ensembles performed similarly.

### 4.3. Limitations

Nonetheless, our system has some limitations. For instance, there are some patients with heterogeneous distributions of cysts all over the abdomen. This makes it difficult to distinguish kidneys from other organs. In such cases, our models over-segment the kidneys by delineating parts of other abdominal organs that also contain cysts (e.g., liver). However, even though under- and over segmentation occur, the differences in the obtained versus manual segmented TKVs is 5% on average throughout all models (cf. Table 3). Furthermore, we have not yet demonstrated generalization in terms of applications/transfer to other domains. We are currently looking into exploiting publicly available data e.g., KiTS19 [41]. Initial results look promising, but in-depth evaluation is pending.

## 5. Conclusions

In this paper, we proposed approaches to overcome the problem of limited data for training convolutional neural networks for segmenting the TKV in kidneys with ADPKD. We demonstrated that combining the cosine loss function and SAM could achieve high segmentation accuracy while only using 100 datasets. Furthermore, TKV was obtained at high accuracy compared to manual segmentation (surpassing the inter-user agreement). Our study shows that fast and automated segmentation and TKV estimation is possible, allowing for clinical translation in the future.

Subsequently, we plan to transfer our methods to the available T2-weighted MR images and investigate a combination of both MR contrasts to fully exploit the benefit of the complementary image information.

## Figures and Tables

**Figure 1 diagnostics-12-01159-f001:**
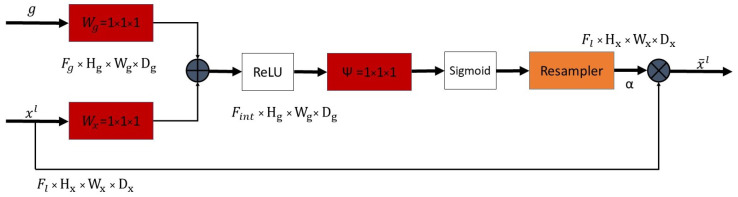
Attention module used in the Attention U-Net. It takes two inputs: *g* and $x^{l}$ where *g* is the higher-level feature and spatially smaller than the previous layer feature $x^{l}$. *g* is used to guide lower-level features $x^{l}$ to emphasize relevant image regions. This is achieved by calculating α coefficients that are element-wise multiplied with $x^{l}$ to produce attention maps. Adapted from [21].

**Figure 2 diagnostics-12-01159-f002:**
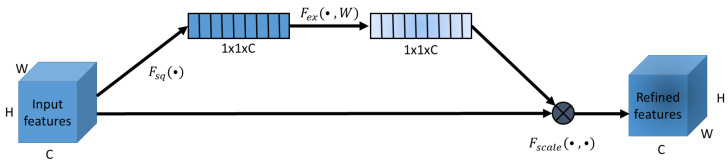
Squeeze and excitation module. The module first squeezes the input features to a vector with the shape 1xC (C: number of channels in the input features). This is followed by the excitation operation where a multi-layer perceptron processes the input vector. The final output feature map is calculated by multiplication between the output vector and the input feature map. Adapted with permission from Ref. [22]. 2018, IEEE.

**Figure 3 diagnostics-12-01159-f003:**
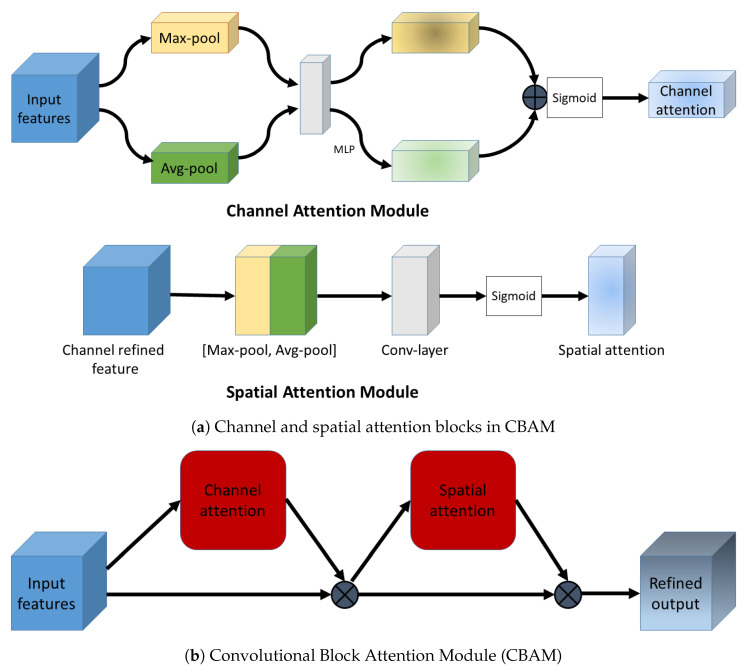
The scheme of the convolutional block attention module. (**a**) The channel attention module implements max and average pooling along the channel dimension. It produces a refined 1D channel attention vector. Meanwhile, the spatial attention module applies max and average pooling along spatial dimensions to produce a 2D spatial attention map. (**b**) The CBAM combines the channel and spatial attention maps (red-colored squares) detailed in (**a**) and applies them to the input feature map sequentially to produce an output feature map, which is refined along both the channel and spatial dimensions. Adapted with permission from Ref. [23]. 2018, Springer Nature.

**Figure 4 diagnostics-12-01159-f004:**
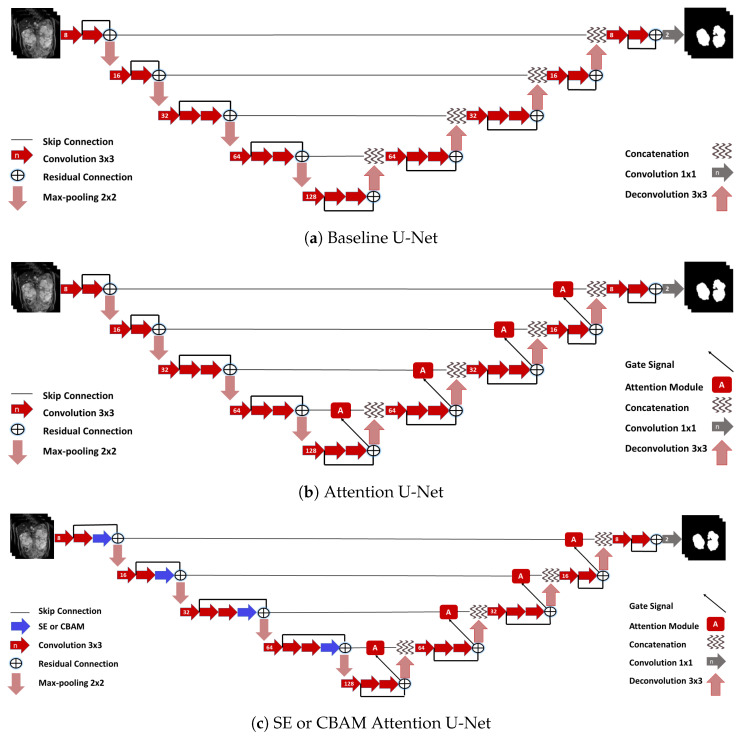
The network architectures for kidney segmentation: (**a**) baseline U-Net without any attention modules, (**b**) U-Net with attention modules as described by Oktay et al. [21] and (**c**) U-Net combining SE or CBAM [22,23] with attention modules from [21].

**Figure 5 diagnostics-12-01159-f005:**
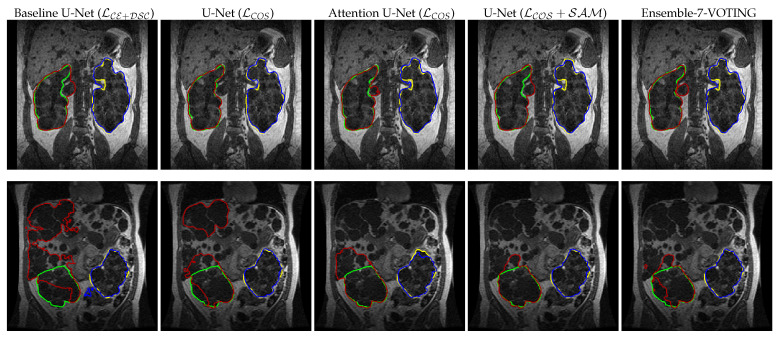
Qualitative results after post-processing (128 x 128): random slices of the best (top row) and the worst case segmentation (bottom row) from the baseline U-Net ($L_{\mathrm{CE}+\mathrm{DSC}}$) compared to its corresponding U-Net ($L_{\mathrm{COS}}$ & $L_{\mathrm{COS}}+\mathrm{SAM}$), Attention U-Net ($L_{\mathrm{CE}+\mathrm{DSC}}$), and Ensemble-7-VOTING predictions. The best case has stage 1 CKD, while, the worst case has stage 2 CKD. The ground truth segmentation is colored in green and yellow, while the network segmentations are colored in red and blue. Top row DSC (left to right): Baseline U-Net = 0.963, U-Net ($L_{\mathrm{COS}}$) = 0.965, Attention U-Net ($L_{\mathrm{CE}+\mathrm{DSC}}$) = 0.965, U-Net ($L_{\mathrm{COS}}+\mathrm{SAM}$) = 0.962 and Ensemble-7-VOTING = 0.969. Bottom row DSC: Baseline U-Net = 0.606, U-Net ($L_{\mathrm{COS}}$) = 0.713, Attention U-Net ($L_{\mathrm{CE}+\mathrm{DSC}}$) = 0.747, U-Net ($L_{\mathrm{COS}}+\mathrm{SAM}$) = 0.784 and Ensemble-7-VOTING = 0.780. There is no significant difference between the baseline and our modified U-Nets in the case of the best segmentation. However, there is a maximum of 18% improvement in the DSC for the worst case. The worst case has cysts all over the abdomen region, which makes model prediction difficult, nonetheless, the attention mechanisms, cosine loss, SAM, and ensembles help improve the segmentation and can be useful in locating cysts in other regions as well.

**Table 1 diagnostics-12-01159-t001:** The DSC and MSSD (in mm) values for various networks with loss functions as DSC ($L_{DSC}$), cross-entropy+DSC ($L_{CE+DSC}$), and cosine loss ($L_{COS}$). The experiments were performed for two patch sizes: 96 and 128, with the numbers in bold implying significant difference (*p*-value < 0.05) between the baselines and the corresponding network configuration. The underlined numbers signify the best in the respective category excluding ensemble results. As can be seen, adding attention to the U-Net can improve the results significantly (Attention and CBAM U-Nets). Furthermore, cosine loss alone (U-Net) or with Attention and CBAM U-Nets provides better DSC than the corresponding cross-entropy+DSC loss networks. Finally, the ensembles outperform the best model in each category. The ensembles in italics imply significantly better results (*p*-value < 0.05) than the corresponding best performing model.

Architecture	Loss	DSC ↑	MSSD (mm) ↓	DSC ↑	MSSD (mm) ↓
		96 × 96	96 × 96	128 × 128	128 × 128
Baseline U-Net	$L_{DSC}$	0.789 ± 0.109	8.803 ± 5.947	0.855 ± 0.079	5.435 ± 5.257
	$L_{CE+DSC}$	0.865 ± 0.075	4.229 ± 4.094	0.888 ± 0.067	3.297 ± 3.951
SE U-Net	$L_{CE+DSC}$	**0.889 ± 0.060**	**3.199 ± 3.941**	0.892 ± 0.060	2.805 ± 3.087
	$L_{COS}$	**0.879 ± 0.080**	3.566 ± 4.164	0.892 ± 0.057	**2.654 ± 2.843**
	$L_{CE+DSC}+SAM$	**0.895 ± 0.060**	**2.357 ± 2.790**	**0.903 ± 0.052**	**2.248 ± 2.719**
	$L_{COS}+SAM$	**0.902 ± 0.055**	**2.228 ± 3.045**	**0.899 ± 0.058**	**2.450 ± 3.272**
CBAM U-Net	$L_{CE+DSC}$	**0.878 ± 0.080**	3.517 ± 4.327	**0.899 ± 0.056**	**2.490 ± 3.013**
	$L_{COS}$	**0.894 ± 0.056**	**2.636 ± 2.648**	**0.898 ± 0.057**	**2.332 ± 2.568**
	$L_{CE+DSC}+SAM$	**0.880 ± 0.064**	3.689 ± 4.120	**0.903 ± 0.060**	**2.549 ± 4.997**
	$L_{COS}+SAM$	**0.885 ± 0.073**	3.520 ± 5.274	**0.902 ± 0.056**	**2.090 ± 2.683**
Attention U-Net	$L_{CE+DSC}$	**0.882 ± 0.069**	**3.331 ± 3.681**	**0.898 ± 0.057**	3.158 ± 4.191
	$L_{COS}$	**0.892 ± 0.060**	**3.382 ± 4.790**	**0.901 ± 0.061**	**2.427 ± 3.250**
	$L_{CE+DSC}+SAM$	**0.886 ± 0.068**	**3.144 ± 4.266**	**0.903 ± 0.054**	**2.461 ± 2.717**
	$L_{COS}+SAM$	**0.896 ± 0.060**	**2.621 ± 3.270**	**0.907 ± 0.057**	**2.338 ± 3.987**
U-Net	$L_{COS}$	**0.885 ± 0.069**	**3.007 ± 3.317**	**0.902 ± 0.061**	**2.228 ± 2.856**
	$L_{COS}+SAM$	**0.899 ± 0.056**	**2.754 ± 3.541**	**0.909 ± 0.049**	**2.417 ± 3.542**
Ensemble-4-STAPLE	$L_{COS}+SAM$	**0.904 ± 0.058**	**2.479 ± 3.621**	** *0.913 ± 0.052* **	** *1.967 ± 2.841* **
Ensemble-7-STAPLE	$L_{CE+DSC}+SAM+L_{COS}+SAM$	**0.903 ± 0.059**	**2.472 ± 3.575**	** *0.916 ± 0.052* **	** *1.732 ± 2.467* **
Ensemble-4-VOTING	$L_{COS}+SAM$	** *0.910 ± 0.051* **	** *1.886 ± 2.615* **	**0.914 ± 0.049**	** *1.506 ± 2.018* **
Ensemble-7-VOTING	$L_{CE+DSC}+SAM+L_{COS}+SAM$	** *0.910 ± 0.051* **	** *1.934 ± 2.690* **	** *0.918 ± 0.048* **	** *1.484 ± 2.083* **

**Table 2 diagnostics-12-01159-t002:** Results after post-processing with the largest-connected components. The two evaluation metrics are the DSC and the MSSD with the bold values being significantly better (*p*-value < 0.05) than the corresponding baseline network. The underlined values represent the best outcomes in their respective category excluding ensemble results. Meanwhile, the highlighted values in yellow imply a significantly better (*p*-value < 0.05) score than the results from the corresponding networks without post-processing from Table 1. The ensembles in italics signify significantly better results (*p*-value < 0.05) than the corresponding best performing model.

Architecture	Loss	DSC ↑	MSSD (mm) ↓	DSC ↑	MSSD (mm) ↓
		96 × 96	96 × 96	128 × 128	128 × 128
Baseline U-Net	$L_{CE+DSC}$	0.880 ± 0.071	2.382 ± 2.950	0.902 ± 0.600	1.530 ± 2.542
SE U-Net	$L_{CE+DSC}$	**0.897 ± 0.058**	**1.687 ± 1.970**	0.899 ± 0.055	1.630 ± 2.054
	$L_{COS}$	0.884 ± 0.088	2.002 ± 3.000	0.899 ± 0.051	1.579 ± 1.791
	$L_{CE+DSC}+SAM$	**0.894 ± 0.070**	**1.580 ± 1.941**	0.904 ± 0.057	1.290 ± 1.225
	$L_{COS}+SAM$	**0.902 ± 0.060**	**1.438 ± 1.600**	0.900 ± 0.079	1.593 ± 2.912
CBAM U-Net	$L_{CE+DSC}$	**0.890 ± 0.070**	**1.937 ± 2.620**	0.906 ± 0.052	1.395 ± 1.480
	$L_{COS}$	**0.901 ± 0.057**	**1.367 ± 1.158**	0.903 ± 0.058	1.504 ± 1.863
	$L_{CE+DSC}+SAM$	**0.892 ± 0.059**	**1.947 ± 2.228**	**0.910 ± 0.054**	1.294 ± 1.644
	$L_{COS}+SAM$	**0.896 ± 0.065**	**1.926 ± 2.717**	**0.908 ± 0.054**	1.346 ± 1.705
Attention U-Net	$L_{CE+DSC}$	0.891 ± 0.076	**1.794 ± 2.046**	**0.910 ± 0.042**	1.371 ± 1.297
	$L_{COS}$	**0.904 ± 0.056**	**1.588 ± 1.912**	**0.910 ± 0.052**	1.377 ± 1.671
	$L_{CE+DSC}+SAM$	**0.895 ± 0.065**	**1.922 ± 2.757**	**0.911 ± 0.051**	1.398 ± 1.852
	$L_{COS}+SAM$	**0.905 ± 0.056**	**1.470 ± 1.713**	**0.913 ± 0.051**	1.312 ± 1.708
U-Net	$L_{COS}$	**0.896 ± 0.074**	**1.812 ± 2.643**	**0.909 ± 0.057**	1.358 ± 1.970
	$L_{COS}+SAM$	**0.908 ± 0.054**	**1.480 ± 1.885**	**0.918 ± 0.044**	**1.199 ± 1.525**
Ensemble-4-STAPLE	$L_{COS}+SAM$	**0.909 ± 0.055**	**1.560 ± 2.087**	**0.919 ± 0.048**	**1.204 ± 1.508**
Ensemble-7-STAPLE	$L_{CE+DSC}+SAM+L_{COS}+SAM$	**0.909 ± 0.054**	**1.534 ± 1.881**	**0.921 ± 0.048**	**1.174 ± 1.528**
Ensemble-4-VOTING	$L_{COS}+SAM$	** *0.914 ± 0.053* **	** *1.204 ± 1.367* **	**0.917 ± 0.048**	**1.125 ± 1.340**
Ensemble-7-VOTING	$L_{CE+DSC}+SAM+L_{COS}+SAM$	** *0.914 ± 0.052* **	** *1.299 ± 1.620* **	** *0.922 ± 0.047* **	**1.094 ± 1.376**

**Table 3 diagnostics-12-01159-t003:** The R${}^{2}$ and mean TKV difference (%) of selected network configurations (post-processed, patch size: 128) for ground truth TKV v/s predicted TKV (ml). The baseline linear fit has R${}^{2}$ value of 0.915. The U-Net with $L_{COS}$ and $L_{\mathrm{COS}}+\mathrm{SAM}$ have R${}^{2}$ values of 0.914 and 0.958, respectively. The Attention U-Net with $L_{\mathrm{COS}}$ and $L_{\mathrm{COS}}+\mathrm{SAM}$ have R${}^{2}$ values of 0.946 and 0.951, respectively. Among the Ensembles, the highest R${}^{2}$ is 0.957. Meanwhile, the Attention U-Net with $L_{\mathrm{CE}}+\mathrm{DSC}$ achieves the overall highest R${}^{2}$ value of 0.962.

Architecture	Loss	R${}^{2}$ ↑	Mean TKV Difference (%) ↓
Baseline U-Net	$L_{CE+DSC}$	0.915	−4.43 ± 18.90
Attention U-Net	$L_{CE+DSC}$	0.962	−2.04 ± 12.20
	$L_{COS}$	0.946	−1.94 ± 16.14
	$L_{COS}+SAM$	0.951	−1.63 ± 15.73
U-Net	$L_{COS}$	0.914	0.16 ± 16.79
	$L_{COS}+SAM$	0.958	−1.72 ± 12.50
Ensemble-4-STAPLE	$L_{COS}+SAM$	0.953	−3.11 ± 14.63
Ensemble-7-STAPLE	$L_{CE+DSC}+SAM+L_{COS}+SAM$	0.957	−4.00 ± 14.82
Ensemble-7-VOTING	$L_{CE+DSC}+SAM+L_{COS}+SAM$	0.952	−0.65 ± 13.76

## Data Availability

Restrictions apply to the availability of these data. The data for this research was obtained from NIDDK at https://repository.niddk.nih.gov/studies/crisp1/ (accessed on 15 March 2021).

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
