# Peer review of "Deep Learning-Based Total Kidney Volume Segmentation in Autosomal Dominant Polycystic Kidney Disease Using Attention, Cosine Loss, and Sharpness Aware Minimization"

_diagnostics, 2022, doi:10.3390/diagnostics12051159_

Round 1
Reviewer 1 Report
The contributions of the work are not clear. It should be listed in the Introduction section. In addition, in L65, the authors' claim that the cosine loss function is not studied before, however, a similar loss function is already introduced in Rethinking Semantic Segmentation: A Prototype View, which should be discussed in the paper.
Attention modules are widely studied in recent segmentation methods like Matnet: Motion-attentive transition network for zero-shot video object segmentation and Group-Wise Learning for Weakly Supervised Semantic Segmentation. These works should be included for a more inclusive review.
The method is only evaluated on a private dataset. The results will be more convinced to evaluate the method on public CT datasets like KiTS.
What's the difference between baseline unet and unet in Table 1 and 2?
Reviewer 2 Report
This work presents an automated method to segment the kidney volumes, implemented by three attention mechanisms into the U-Net, which aims to avoid the scarcity of large annotated datasets. Overall, the paper is well written, and the experiments are well designed. However, some issues should be considered:
- In fig.1, please clarify the symbol ‘x.’ If it is ‘\times,’ please use the mathematical symbol ‘\times.’
- Unclear what do you want to say by the highlight in table 2.
- Please include the limitation in the section of the discussion.
- “We demonstrated that combining the cosine loss function and SAM could achieve high 353 segmentation accuracy while only using 100 datasets.”. Which one of the two is important to improve segmentation accuracy? Please include more discussion.
- In eq.1, what is the definition by \mu and \theta?
- The information from the Fig.5 is minimal since it is hard to see if there are differences across sub-figures.
Round 2
Reviewer 1 Report
The revision has well addressed my concerns.